# *Citrus bergamia*: Kinetics of Antimicrobial Activity on Clinical Isolates

**DOI:** 10.3390/antibiotics11030361

**Published:** 2022-03-08

**Authors:** Angela Quirino, Valeria Giorgi, Ernesto Palma, Nadia Marascio, Paola Morelli, Angelo Maletta, Francesca Divenuto, Giuseppe De Angelis, Valentina Tancrè, Saverio Nucera, Micaela Gliozzi, Vincenzo Musolino, Cristina Carresi, Vincenzo Mollace, Maria Carla Liberto, Giovanni Matera

**Affiliations:** 1Department of Health Sciences, Institute of Clinical Microbiology, “Magna Graecia” University of Catanzaro, 88100 Catanzaro, Italy; quirino@unicz.it (A.Q.); valeg86@yahoo.it (V.G.); nmarascio@unicz.it (N.M.); morellipaola1@gmail.com (P.M.); angelo.maletta@studenti.unicz.it (A.M.); francydi88@hotmail.it (F.D.); giuseppe.dea@live.it (G.D.A.); valentina.tancre@virgilio.it (V.T.); mliberto@unicz.it (M.C.L.); mmatera@unicz.it (G.M.); 2Institute of Research for Food Safety & Health (IRC-FSH), “Magna Graecia” University of Catanzaro, 88100 Catanzaro, Italy; palma@unicz.it (E.P.); saverio.nucera@hotmail.it (S.N.); micaela.gliozzi@gmail.com (M.G.); v.musolino@unicz.it (V.M.); mollace@libero.it (V.M.)

**Keywords:** *Citrus bergamia*, antimicrobial activity, multi-drug-resistant bacteria, *time-kill assay*, fungicidal activity

## Abstract

Background: The inappropriate use of antibiotics has increased selective pressure and the spread of multi-drug-resistant (MDR) pathogens, which reduces the possibility of effective treatment. A potential alternative therapeutic approach may be represented by essential oils, such as the distilled extract of bergamot (*Citrus bergamia* Risso et Poiteau). Such natural products exercise numerous biological activities, including antimicrobial effects. Methods: This work aimed to evaluate the kinetics of the bactericidal and fungicidal activity of the distilled extract of bergamot on MDR bacteria and fungi from clinical specimens using the *time-kill assay*. Furthermore, the antimicrobial activity of the distilled extract of bergamot on the morphology and cellular organization of clinical pathogens was evaluated by confocal laser scanning microscopy. Results: Our results demonstrated that the distilled extract of bergamot exhibited significant antimicrobial activity and a specific bactericidal effect against the bacterial and fungal strains tested. Furthermore, confocal microscope images clearly showed compromised membrane integrity, damage and cell death in bacterial samples treated with the distilled extract of bergamot. In addition, progressive alterations in cell-wall composition, cytoplasmic material and nucleus structure triggered by exposure to the distilled extract of bergamot were identified in the fungal samples considered. Conclusions: Our data suggest that the use of essential oils, such as distilled extract of bergamot (*Citrus bergamia* Risso et Poiteau), can represent a valid alternative therapeutic strategy to counteract antibiotic resistance of pathogens.

## 1. Introduction

The continued and often inappropriate use of antibiotics has increased selective pressure, favoring the emergence, growth and spread of resistant microorganism strains [1]. The appearance of multiple antibiotic-resistant pathogens (multi-drug Resistance, MDR) further reduces the possibility of effective treatment. The World Health Organization (WHO) reported a list of MDR microorganisms responsible for high mortality, especially in healthcare settings [2]. Among the WHO priority pathogens, methicillin-resistant *Staphylococcus aureus* (MRSA) recently developed resistance to three widely used antibiotics (linezolid, vancomycin, and daptomycin), forming a drug-tolerant biofilm [3]. Few antimicrobial drugs are effective against carbapenem-resistant *Acinetobacter baumannii* (CRAB), classified as critical by the WHO, such as ampicillin-sulbactam in monotherapy or in combination [4]. The prevalence of carbapenem-resistant *Klebsiella pneumoniae* (CRKP) and ceftazidime/avibactam (CZA) resistance is increasing, with a negative impact on nosocomial infection [5]. Finally, Candida species (the most common opportunistic pathogen) are increasingly resistant to azole drugs with potential fungicidal effects [6]. Therefore, the therapeutic challenge to treating these infections requires the development of novel antimicrobial agents. A potential alternative therapeutic approach for the treatment of some infectious diseases can be represented by essential oils, which, with their complex and varied chemical composition, exercise a number of biological activities [7,8,9].

Several studies showed the in vitro antimicrobial properties of bergamot (*Citrus bergamia* Risso et Poiteau) on several microorganisms, including Gram-positive and Gram-negative bacteria and yeasts. In particular, the distilled extract can contrast emerging pathogens responsible for healthcare-related infections that are increasingly difficult to control and treat. Scientific interest also turned to evaluating the potential of the individual components of bergamot distilled extract [10]. Quirino et al. [11] highlighted the bactericidal and fungicidal activity of this natural substance using plate microdilution assays. The results showed that the distilled bergamot extract had a significant effect on all microorganisms tested [11]. In a recent study, the major components of the distilled extract of bergamot (limonene, linalool, linalyl acetate, and γ-terpinene) were tested both individually and in combination on a wide range of bacteria and Candida species. From in vitro testing, it emerged that the best antibacterial and antifungal capacity was given by the synergism between the components rather than by a single molecule [12].

The aim of this work was to evaluate the kinetics of the bactericidal and fungicidal activity of the distilled extract of bergamot on *multi-drug-resistant* bacteria and fungi of clinical isolates using the *time-kill assay*. The morphology and cellular organization of clinical isolates were evaluated by confocal laser scanning microscopy to understand the possible antimicrobial mechanisms of the distilled extract of bergamot.

## 2. Materials and Methods

### 2.1. Citrus Bergamia Risso et Poiteau

*Citrus bergamia* Risso et Poiteau belongs to the family *Rutaceae*, genus *Citrus* and it grows almost exclusively in Calabria, Italy, along the southern east coast in the area of Reggio Calabria. Bergamot is a small evergreen tree that, in cultivation, can grow up to 5 m in height, with an erect, cylindrical, dark grayish-brown stem and irregular branches (Figure 1). The leaves have a short, sometimes winged petiole and are persistent, simple, alternate and aromatic when bruised. The lamina is ovate-oblong or lanceolate, usually with a sharp apex and a crenated margin or slightly wavy, hairless, and leathery, with a dark-green upper surface and light-green lower surface. The inflorescence is racemose and many-flowered. The flowers are fragrant, actinomorphic and pentamerous. The calyx persists in the ripened fruit. The corolla often presents five white petals inserted alternately with the sepals. Oil glands are also present in the petals. The hesperidium (*Citrus* spp. fruit) is slightly flattened and subglobose to pyriform. The peel is 3–6 mm thick, with numerous glands containing the essential oil, smooth to rough, adherent and shiny green turning to pale yellow when the fruit is ripe. The mesocarp is white, whilst the endocarp is divided into 10 to 15 lodges containing a sour and bitter greenish-yellow juice.

### 2.2. Preparation of Distilled Extract of Bergamot

The distilled extract of bergamot used in this study was a kind gift by *Capua S.r.l.*, Reggio Calabria, Italy. Briefly, according to the manufacturer’s instructions, bergamots were peeled, and the peel was chopped and minced thoroughly. The cold expressed essential oil obtained was collected and subjected to vacuum distillation. This process involves heating the original bergamot essential oil and vaporizing the compounds, which depends on the volatility of the components. The packaging type used was “rashing rings”. The distillation temperature ranged between 20 °C and 180 °C, while the vacuum range was between 0.05 and 15 millibars.

### 2.3. Chemical Characterization of the Distilled Extract of Bergamot

The chemical characterization of the distilled extract of bergamot was performed by the Hewlett Packard (Agilent, Santa Clara, CA, USA) gas chromatograph (model HP 5890A)—mass spectrometer (model HP 5972A) equipped with an HP-35MS column. Helium was used as the carrier gas, and ionization was obtained by electron impact. Column temperature was maintained at 60 °C for 5 min and then raised to 280 °C in 10 °C/minute increments. Nineteen compounds were identified and quantified. Other compounds were identified by comparison of mass spectra of each peak with those of authentic samples in the NIST (National Institute of Standard and Technology) library. The qualitative and quantitative gas-chromatographic analysis of the distilled extract of bergamot showed a high level of some major aromatic compounds, as shown in Table 1.

### 2.4. Microorganisms Tested

Thirty-one *multi-drug-resistant* microorganisms isolated from clinical samples of patients admitted to the “Magna Græcia” University Hospital of Catanzaro (Italy) were selected (Table 2).

The identification and sensitivity to antibiotics of the microorganisms tested were evaluated with the automated system Vitek 2 (BioMérieux, France) according to the European Committee on Antimicrobial Susceptibility Testing (EUCAST) guidelines. Antimicrobial activities of the distilled extract of bergamot were evaluated using different methods such as a micro-well dilution assay and confocal microscopy. Moreover, the time-kill assay was carried out to evaluate the kinetics of the bactericidal and fungicidal activity of the distilled extract of bergamot on isolates.

### 2.5. Broth Microdilution Assay

Bactericidal and fungicidal activity was analyzed using broth microdilution assays followed by the determination of the minimum bactericidal concentration (MBC) as previously described by Quirino et al. [12]. Aliquots of 1 mL of distilled extract of bergamot were placed in sterile glass test tubes and emulsified with Tween 20, an inert, non-ionic tensioactive agent with no inherent bactericidal and fungicidal activity (900 μL of substance plus 100 μL of Tween 20). The mixture was then vortexed, and the first dilution (1:20) of the substance was performed in a liquid medium (Nutrient Broth, BioMérieux). In total, 100 μL of the first dilution was dispensed into the first well of a sterile microtitration plate, and two-fold serial dilutions were then performed. Microbial suspensions were adjusted to a final concentration of 1 × 10^6^ CFU/mL and distributed in each well of a 96-well plate. In each well, a wide range of concentrations, from 5% to 0.03% (*v*/*v*), was obtained by the two-fold serial dilution of the investigated substance. The microplate was incubated for 24 h at 37 °C. To determine the antimicrobial effect of distilled bergamot extract, subcultures of 1 µL from each dilution were inoculated onto blood agar plates (bioMérieux, Craponne, France) and incubated for 24 h at 37 °C in aerobic conditions, before colony count. The first dilution of the substance at which no microbial growth was observed represents the minimal bactericidal concentration (MBC).

All experiments included the substance alone to exclude possible contamination (negative control), while bacterial suspension without substance was used as a control sample of microbial growth. Each experiment was performed in triplicate.

### 2.6. Time-Kill Assay

Data are presented in terms of log_10_ CFU/mL. The bactericidal effect of a substance, such as the distilled extract of bergamot, is defined as the 3 log_10_ CFU/mL reduction of microbial colonies compared to the initial inoculum. A reduction of less than 3 log_10_ CFU/mL is defined as a bacteriostatic effect [13].

A volume of 100 µL of bergamot distilled extract, at a concentration of 1 × MBC for each microorganism used, was dispensed in the wells of a 96-microwell plate. Microbial cultures in exponential growth were diluted in nutrient broth (bioMérieux, France) up to the concentration of 1 × 10^5^ CFU/mL, and 100 µL were dispensed in each well of the microplate. Subsequently, the microplate was incubated at 37 °C and, at pre-established time intervals (0, 30′, 1, 2, 4, 6, 8, 24 h), a 1 µL volume was inoculated onto blood agar plates (bioMérieux, France). After incubation at 37 °C for 24 h, colonies on individual plates were counted and expressed as the number of colony forming units/mL (CFU/mL).

All experiments were performed in triplicate. The decrease in growth was compared with a growth control of the untreated strain. The killing rate was determined by plotting the logarithm of the viable counts (CFU/mL) against time.

### 2.7. Confocal Microscopy

The interaction of the distilled extract of bergamot on multi-drug-resistant (MDR) bacteria and *Candida* cells was investigated using confocal laser scanning microscopy. MDR bacteria were observed under confocal microscopy during incubation with the distilled bergamot extract at a concentration of 1 × MBC. In particular, microscopic preparations of untreated strains and strains treated with the substance after at least 4 h of incubation were prepared. At different times, each preparation was washed with PBS buffer and fixed in formalin. To determine the viability of bacterial cells after treatment, as well as the bactericidal activity of the distilled extract of bergamot, 10 µL of each preparation was labeled with 10 µL of acridine orange (Sigma Aldrich, St. Louis, MO, USA), a nucleic acid selective metachromatic stain. Following the acridine orange procedure, differential staining was observed based on the presence of live or dead cells. Indeed, live cells appeared green while dead cells were stained red. Images were acquired with a 63X objective using the *Leica TCS SP5* Confocal Microscopy System (Leica Microsystems, Wetzlar, Germany).

Each *Candida* strain was tested with a 1 × MBC of distilled extract of bergamot at three different times: immediately after the addition of the substance, at the mean time of growth reduction and at the time indicated as no growth. At the selected times, cells treated with distilled bergamot extract were washed with PBS buffer and fixed in formalin. For the determination of cell wall components, 200 µL of the microorganism suspensions were labeled with two different dyes: 50 µL of Concanavalin A (ConcA, Sigma Aldrich) and 30 µg/mL of Calcofluor White (CFW, Sigma Aldrich). ConcA selectively binds to α-mannanopyranosyl and α-glucopyranosyl residues; CFW is a non-specific fluorochrome that binds cellulose and chitin. Furthermore, 15 µL of DAPI stain (4′,6-diamidine-2-phenylindole, Sigma Aldrich) was used to evaluate the nuclear morphology. After 15 min for the DAPI stain and 60 min for the ConcA and CFW stains of incubation in a dark chamber, each specimen was observed under the confocal microscope.

One slide untreated with the substance and one unlabeled slide were added to each observation to exclude background interference or autofluorescence. Each experiment was performed in duplicate. The protocol described for the present work was adapted from Xing et al. [14].

## 3. Results

### 3.1. Antimicrobial Activity

The antimicrobial activity of the distilled extract of bergamot was determined by MBC values against 31 tested bacterial and fungal strains. Minimal bactericidal concentration is defined as 99.9%, or greater, killing efficacy at a selected time (Table 3).

The microorganisms tested showed different sensitivity to the distilled extract of bergamot. A first difference was noted within the bacteria: the bactericidal effect in Gram-positive bacteria was observed only at high concentrations of the substance (5–2.5% *v*/*v*) compared to concentrations (2.5–0.625% *v*/*v*) needed to kill Gram-negative bacteria.

The results obtained from tests on *Candida* species showed more significant heterogeneity: one isolate was sensitive to a distillate concentration of 0.313% *v*/*v*, while complete growth inhibition in the other isolates required distillate concentrations greater than 1.25% *v*/*v*.

### 3.2. Time-Kill Assay

The bactericidal effect on the Gram-positive bacteria was obtained only at high concentrations of distilled bergamot extract (5% and 2.5% *v*/*v*) and at the longest exposure times. In the first 6 h of incubation, no significant reduction in the number of log_10_ CFU/mL was observed in the three *S. aureus* strains (Figure 2A) showing a bactericidal effect only after 8 h. *S. epidermidis* and *S. haemolyticus* showed a similar pattern to *S. aureus.* but the bactericidal effect occurred after a longer time of exposure (between 8 and 24 h) (Figure 2B).

More interesting results were obtained on Gram-negative bacteria. Ten strains of *A. baumannii complex* were monitored over 24 h using the kill curves (Figure 2C). In particular, a concentration of 0.625% *v*/*v* of the distilled extract of bergamot caused a bactericidal effect against *A. baumannii 22/19*, with a reduction of 5 log_10_ CFU/mL in just 30 min of exposure. However, in most *A. baumannii* strains tested, the bactericidal effect was observed between 6 to 8 h of exposure. Nine strains of *K. pneumoniae* multi-drug resistance (MDR) were tested (Figure 2D). A steady reduction in microbial growth of 1 log_10_ CFU/mL every hour was observed in *K. pneumoniae 11/19*; while exposure of *K. pneumoniae 24/19* and *K. pneumoniae 23/19* at a concentration of 1.25% (*v*/*v*) of the distilled extract of bergamot was sufficient to cause complete absence of growth after 30 and 60 min, respectively. In the other strains tested, the bactericidal effect was observed between 4 to 8 h of exposure.

The killing tests performed on *Candida* strains showed an inverse correlation between the distilled extract of bergamot concentration and the time of complete inhibition of growth. Indeed, the bactericidal effect on *C. albicans 1/18* and *C. albicans 5/18* was observed after only one hour of exposure with a distillate concentration of 2.5% (*v*/*v*), while on *C. albicans 25/19*, *C. albicans 26/19,* and *C. albicans 27/19*, the FBC value of 1.25% (*v*/*v*) required a longer exposure time to have the fungicidal effect. Similarly, the fungicidal effect on *C. parapsilosis 16/19*, with an FBC value of 1.25% *v*/*v*, was obtained after 2 h (Figure 2E). Finally, *C. glabrata 10/19* was the most sensitive strain to the substance tested, with an FBC of 0.313% (*v*/*v*): the growth curve showed a similar trend to the untreated control for 4 h, and the complete absence of growth occurred after 6 h (Figure 2F).

### 3.3. Confocal Microscopy

Orange acridine staining (Figure 3) showed dead cells after treatment with the distilled bergamot extract for at least 4 h compared to untreated cells. Confocal microscopy confirmed results obtained with the time-kill Assay and also showed a modified cell morphology after 4 h of treatment.

The main components of the cell membrane and the organization of the genetic material of *C. albicans* cells treated with the distilled extract of bergamot were analyzed with confocal laser scanning microscopy and fluorescent dyes, comparing them to those of untreated cells.

Chitin was directly labeled with calcofluor white (CFW) (blue color) and α-mannopyranosys with Concanavalin A (ConcA) (red color). Moreover, the spectral overlay of the two fluorescent dyes permitted simultaneous analysis. Images of *C. albicans 1/18* visualized by confocal microscopy during exposure to distilled extract of bergamot are reported in Figure 4. Evaluation by confocal microscopy did not allow for the precise quantitative assessment of cell wall components, but it did facilitate imaging of changes in cell wall composition triggered by exposure to the distilled extract of bergamot. In cells not treated with the distilled extract of bergamot, there was a strong fluorescent signal that indicates the structural integrity of the cell wall, in particular of the inner layer of chitin and the outer layer of mannan. Within 60 min in the treated cells, the ConcA fluorescence intensity showed a gradual and evident reduction. Nevertheless, the fluorescent signal intensity for CFW seemed to remain stable, and a slight reduction was evident only after 60 min. Through the overlapped images, it is possible to highlight the reorganization of the treated cell wall: CFW and ConcA staining showed a heterogeneous distribution of chitin and mannan, respectively, in the cell wall. In untreated *C. albicans* cells, DAPI staining highlighted the rounded, blue nucleus with finely dispersed chromatin (Figure 5). The cells treated at time 0′, with the antimicrobial substance tested, showed minimal nuclear structural differences from the control cells, but the nucleus was still well defined and stained in blue. Similarly, at the median incubation time, the DAPI still detected the presence of a nuclear structure even if with a lower intensity of fluorescence, indicating lower compactness of the genetic material. Instead, cells treated for up to 60 min showed some distinct morphological differences: the typical rounded structure of the nucleus was no longer observable, while the presence of more fluorescent signals dispersed in the cell’s cytosol was evident. It is interesting to note how progressive alterations of the cell wall and cytoplasmic material of *C. albicans* cells, treated with distilled extract of bergamot, reflect the trend of the killing kinetic curve.

Optical Microscopy of *C. albicans 1/18* clinical isolate treated for up to 60′ with the distilled extract of bergamot showed a significant cellular shrinkage highlighting the antimicrobial activity of bergamot (data not shown).

## 4. Discussion

The emergence of multi-drug-resistant (MDR) pathogens has become a major problem around the world with a significant clinical and economic impact. Within this scenario, the possibility of using essential oils, such as distilled extract of bergamot, represents a valid therapeutic alternative, as our data showed.

Bergamot (*Citrus bergamia* Risso et Poiteau) represents an ancient fruit, historically used for the production of essential oil widely applied to the cosmetic, pharmaceutical and food industries. Furthermore, more recent studies reported that bergamot juice or its enriched polyphenolic fraction (BPF), obtained from the peeled fruit, is mainly composed of flavanones (such as naringenin, hesperetin, eriodictyol glycosides), flavones (apigenin, luteolin, chrysoeriol, diosmetin glycosides) and their 3-hydroxy-3-methyl-glutaryl (HMG) derivatives [15,16,17]. Several in vitro and in vivo experiments employing these compounds clearly showed important hypolipemic, hypoglycemic, anti-inflammatory and cardioprotective effects as well as cholesterol-lowering activity by modulating hepatic HMG-CoA levels, binding bile acids and increasing the turnover rate of blood and liver cholesterol [18,19,20,21,22].

Bergamot essential oil is extracted from the peel and is rich in furocoumarins (bergapten and bergamottin, in particular), coumarins, pectins and flavonoids, which are suggested to exert important health-related properties, especially due to their antioxidant and anti-inflammatory activity [23,24]. Moreover, these compounds show a natural defensive role against invading pathogens, including bacteria, fungi, and viruses [25]. The most abundant compounds, limonene, linalool and linalyl acetate, are responsible for the synergistic effect of antimicrobial activity [12]. The antimicrobial activity of the monoterpenes was previously investigated, and is related to the lipophilicity and water solubility, then significantly influenced by their physicochemical characteristics, and the composition of bacterial membranes [26].

The distilled extract possessed the greatest antimicrobial efficacy against MDR Gram-negative bacteria (such as *Acinetobacter baumannii* and *Klebsiella pneumoniae*) and *Candida albicans* responsible for nosocomial infections representing a worldwide public health problem. On the contrary, Gram-positive bacteria and, in particular, *Staphylococcus* strains are more resistant to hydrophobic molecules due to the presence of slime [27,28]. The mechanism of interaction and entry of the distilled extract into bacterial cells is related to the molecular composition of the external membranes [29,30].

The present study evaluated the kinetics of the bactericidal and fungicidal activity of the distilled extract of bergamot on MDR bacteria and fungi of clinical isolates using the time-kill assay. This in vitro procedure is able to provide dynamic information about the decrease in microbial growth in relation to the time of exposure to the substance.

The bactericidal effect on the Gram-positive bacteria was observed only at a high concentration of distilled extract and after long-time exposure. A similar pattern of killing kinetics was analyzed on clinical MRSA, which demonstrated bactericidal response at 1% (*v*/*v*) and 2% (*v*/*v*) to tea tree oil and 0.12% (*v*/*v*) and 0.25% (*v*/*v*) to thyme oil after 24 h and 48 h of incubation, respectively [31]. Therefore, this data confirms the need for longer incubation times with the natural antimicrobial agent to achieve complete growth inhibition.

Distilled extract of bergamot showed enhanced antimicrobial activity on Gram-negative bacteria. Indeed, the bactericidal effect was observed at shorter exposure times and sometimes with a lower concentration of the substance. Similar killing curves were observed during processing with the essential oil from *Zingiber cassumunar* against the extensively drug-resistant (XDR) *A. baumannii* strains. This essential oil could completely inhibit *A. baumannii* 1 h after treatment [32]. *Z. cassumunar* was also shown to exhibit antibacterial effects against Gram-positive bacteria, such as *S. aureus*, with very low antibacterial activity [33], while it displayed high antimicrobial activity against dermatophytes and yeasts [34]. The antimicrobial activity of *Origanum vulgare* L. essential oil (OVeo) against CRKP, *Serratia marcescens*, and *Acinetobacter baumannii* was assessed by the time-kill assays, and a decrease in cell count was observed after 4 h treatment [35].

García-Salinas et al. evaluated the bactericidal mechanism of molecules derived from different essential oils, such as carvacrol, cinnamaldehyde and thymol, which were shown to be very effective against Gram-negative and Gram-positive bacteria. Indeed, confocal microscope images clearly confirmed the membrane damage exerted by the tested compounds against the bacteria, showing the compromised membrane integrity [36]. Similar antimicrobial findings caused by extracts from *Thymus vulgaris* and *Myrthus communis* were reported [37].

Candida strains, particularly *C. albicans*, were found to be extremely sensitive to the distilled extract of bergamot. Additionally, citral, the main phytoconstituent of other essential oils, such as *Mentha piperita* L. (Briq), *Origanum vulgare* and *Zingiber officinale* L. showed strong antifungal activity against *Candida albicans* strains. In the time-kill curve, the reduction in the growth of clinical strains was equal to 3 log_10_ CFU/mL after exposure of citral for 2 h [38].

In addition, Shanina et al. suggested that not only does the main compound of the substance tested have the ability to alter the integrity of the wall, but also minor compounds such as limonene, linalool and others contribute to the fungicidal action [39].

The confocal laser scanning microscopy is able to highlight the structural changes of the cell membrane, the first defensive barrier against cytotoxic substances, and at the nuclear level. In addition, the use of fluorescent dyes allows specific marking of the cell wall. It will therefore be possible to evaluate the cellular integrity of different components of *Candida albicans* cells and thus observe the extent of cell damage following exposure with distilled extract of bergamot in relation to the different intensity of fluorescence.

Evaluation by confocal microscopy did not allow precise quantitative assessment of cell wall components but facilitated the imaging of changes in cell wall composition triggered by exposure to distilled extract of bergamot. It is interesting to note how the progressive alterations of the cell wall and cytoplasmic material of *C. albicans* cells, treated with distilled extract of bergamot, reflect the trend of the killing kinetic curves. Other essential oils, such as those of *Cinnamomum zeylanicum* Blume bark and leaf, demonstrated fungicidal activities at very low concentrations. The mechanism of action of this oil on *C. albicans* and *C. auris* was damage to the envelope, evident by the shrinkage of the cell surfaces, reducing the cytoplasm, thus leading to cell lysis [40]. Moreover, envelope sterols of *Candida* spp., have a good affinity for different constituents of bergamot, suggesting a potential role for bergamot in topical treatment [11].

## 5. Conclusions

The emergence of multi-drug-resistant (MDR) pathogens has become a major problem worldwide with a significant clinical and economic impact. Within this scenario, the possibility of using essential oils, such as the distilled extract of bergamot (*Citrus bergamia* Risso et Poiteau), represents a valid therapeutic alternative, as our data showed.

Broth microdilution and time-kill assays demonstrated significant antimicrobial activity and a specific bactericidal effect against bacteria and fungi when studied with the distilled extract of bergamot. Confocal microscope images in bacterial samples treated with the natural extract clearly showed compromised membrane integrity, damage and cell death. In particular, in the fungal samples, it triggered progressive alterations in cell wall composition, cytoplasmic material and nucleus structure.

Furthermore, the combined action of the distilled extract of bergamot with antibiotics could represent both a strategy to eradicate antibiotic-resistant microorganisms and the possibility of decreasing the concentrations of antibiotics used, with a consequent reduction in the spread of drug resistance.

## Figures and Tables

**Figure 1 antibiotics-11-00361-f001:**
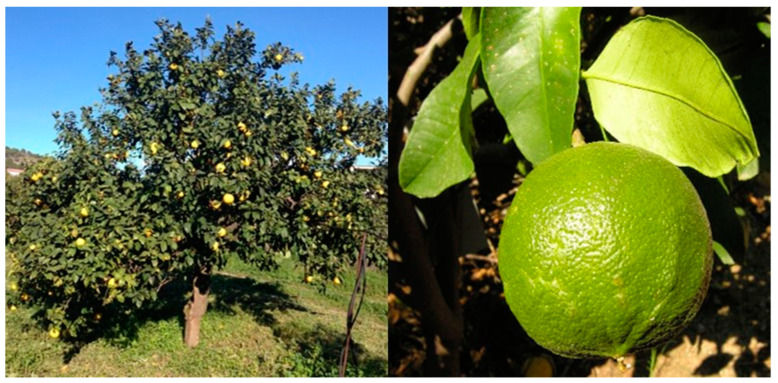
*Citrus bergamia* Risso et Poiteau.

**Figure 2 antibiotics-11-00361-f002:**
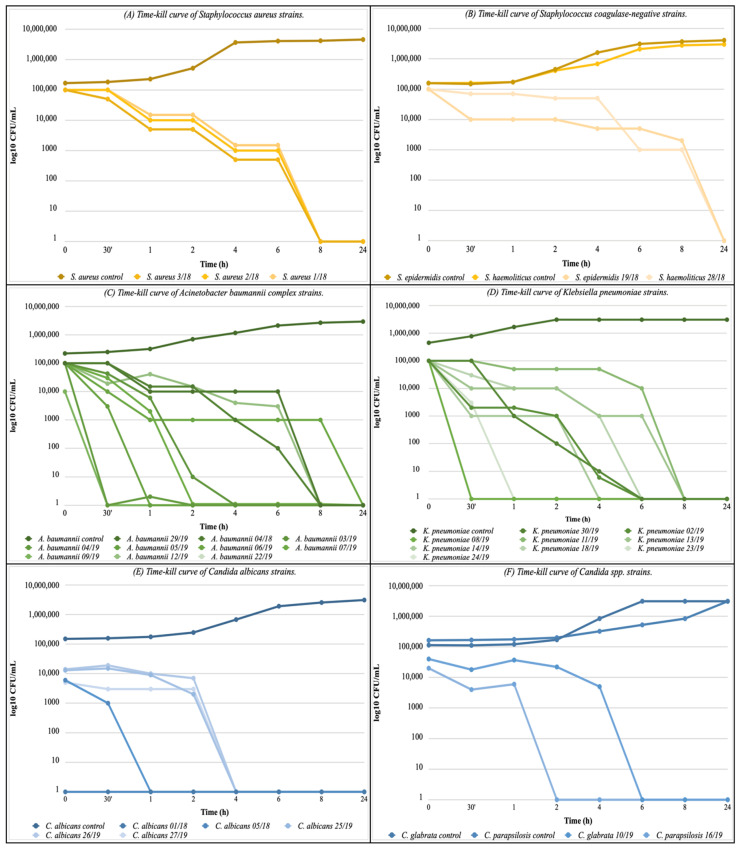
Time-kill curves of distilled extract of bergamot 1 × MBC against: (**A**) *Staphylococcus aureus* strains; (**B**) *Staphylococcus coagulase-negative* strains; (**C**) *Acinetobacter baumannii complex* strains; (**D**) *Klebsiella pneumoniae* strains; (**E**) *Candida albicans* strains; (**F**) *Candida* spp. strains.

**Figure 3 antibiotics-11-00361-f003:**
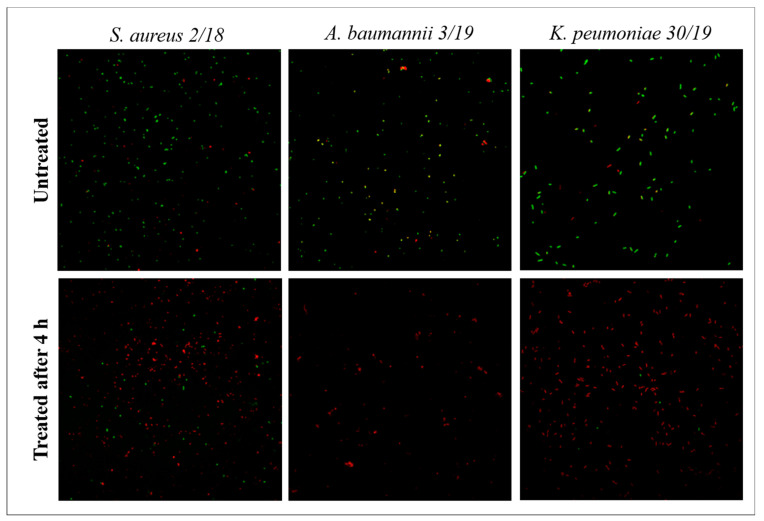
Representative images of confocal laser scanning microscopy of MDR bacteria treated with the distilled extract of bergamot and stained with acridine orange. “Untreated”: control of bacterial growth; “treated”: strains treated with the distilled extract of bergamot (MBC). Green fluorescence shows the presence of living cells; red fluorescence shows the presence of dead cells.

**Figure 4 antibiotics-11-00361-f004:**
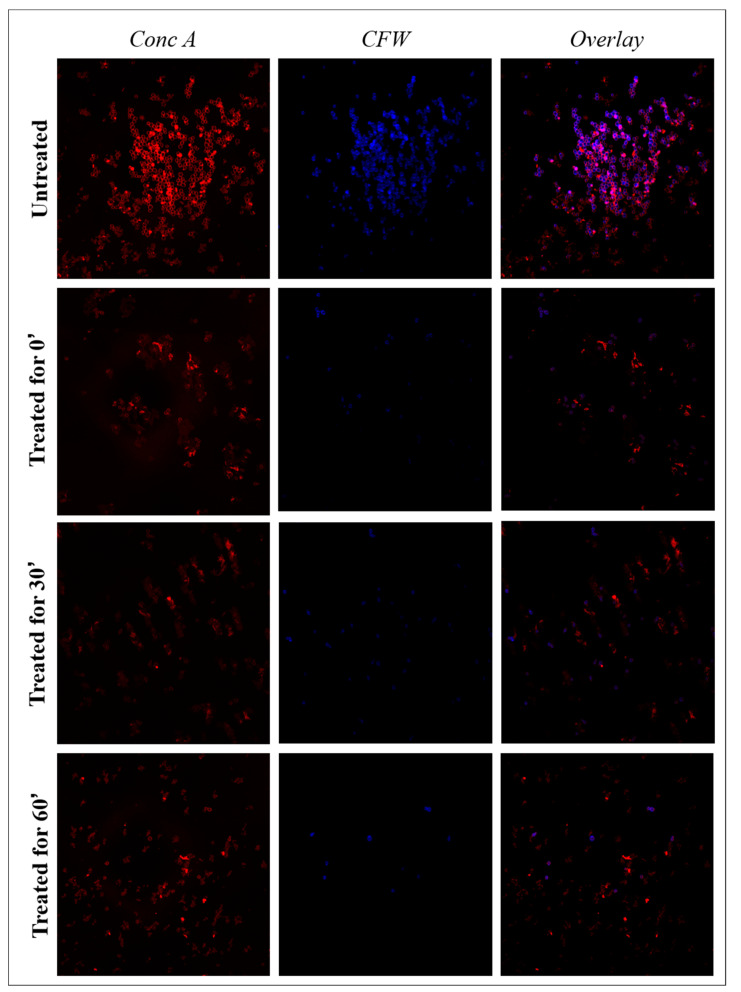
Confocal laser scanning microscopy of cell wall components of *C. albicans 1/18* clinical isolate treated with the distilled extract of bergamot for 60′. Double-staining of chitin and mannan fluorescently labeled with Calcofluor white (CFW) and Concanavalin A (Conc A), respectively, was performed. “Untreated”: growth control of *C. albicans 1/18*; “treated”: *C. albicans 1/18* treated with the distilled extract of bergamot (MBC). The reduction in fluorescence intensity shows a decrease in the relative cell wall components.

**Figure 5 antibiotics-11-00361-f005:**
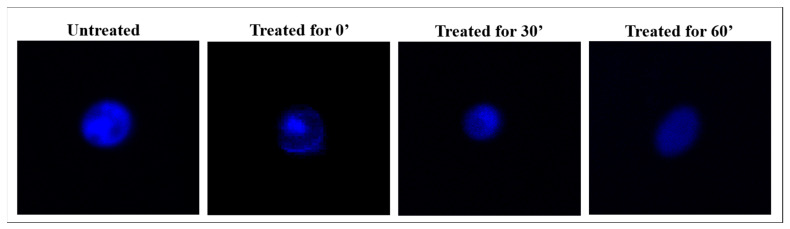
Confocal laser scanning microscopy of *C. albicans 1/18* clinical isolate exposed for up to 60′ to the distilled extract of bergamot. DAPI staining for DNA identification was performed. “Untreated”: growth control of *C. albicans 1/18*; “treated”: *C. albicans 1/18* treated with the distilled extract of bergamot.

**Table 1 antibiotics-11-00361-t001:** Chemical characterization of the distilled extract of bergamot.

Compounds	t_R_ (min.)	Distillate (*w*/*w*%)	Bp (°C)
**Cyclic monoterpenes**			
α-pinene	4.24	1.03	155
β-pinene	5.99	6.56	167–167
α-phellandrene	6.89	0.04	171–172
α-terpinene	7.18	0.16	173–175
Limonene	7.48	30.2	176
p-cimene	7.89	0.18	177
ɣ-terpinene	8.37	11.95	182
Terpinolene	8.99	0.27	184–185
**Acyclic monoterpenes**			
Myrcene	6.29	0.82	165
Ocimene	7.75	0.08	65–66
**Oxygenated acyclic monoterpenes**			
Linalool	9.47	21.82	196–198
Lynalyl acetate	12.03	16.21	220
Neral	12.6	0.21	103
Geranial	13.09	0.11	229
Neryl acetate	13.91	0.28	134 (25 mmHg)
**Oxygenated cyclic monoterpenes**			
Terpineol	11.64	0.87	213–218
**Ester**			
Octyl acetate	11.38	0.10	203–213
**Aldehydes**			
Decanal	11.52	Trace	93–95
**Sesquiterpenes**			
Cariofillene	14.38	0.14	128–129

**Table 2 antibiotics-11-00361-t002:** Isolation site and antibiotic resistance profiles of clinical isolates.

ID Stain	Isolation Site	Susceptible	Resistant
*S. aureus 3/18*	MRSA nasal swab	FA, CM, E, GM, LZD, MUP, RA, TEC, TE, TGC, SXT, VA	P, LVX, OX
*S. aureus 2/18*	MRSA nasal swab	FA, GM, LZD, MUP, TEC, TE, TGC, SXT, VA	CM, E, P, LVX, OX, RA
*S. aureus 1/18*	MRSA nasal swab	FA, CM, E, GM, LZD, MUP, RA, TEC, TE, TGC, SXT, VA	P, LVX, OX
*S. epidermidis 19/19*	Wound swab	LZD, RA, TE, TGC, VA	FA, CM, E, GM, LVX, OX, SXT
*S. haemolyticus 28/19*	Wound swab	CM, LZD, RA, TGC, VA	FA, E, GM, LVX, OX, SXT
*A. baumannii 29/19*	Throat swab	AMC, CS	CTX, MEM, GM, CIP, FOS, FT, SXT
*A. baumannii 04/18*	Throat swab	CS	FOX, CTX, MEM, AM, GM, CIP, FOS, FT, CS, SXT
*K. pneumoniae 30/19*	Rectal swab	GM	AMC, TZP, FOX, CTX, CAZ, FEP, ETP, MEM, AM, CIP, FOS, FT, CS, SXT
*K. pneumoniae 02/19*	Rectal swab	AMC, TZP, ETP, MEM, AM, GM, FOS, CS	CTX, CAZ, FEP, CIP, SXT
*A. baumannii 03/19*	Wound swab	CS	MEM, AM, GM, CIP, SXT
*A. baumannii 04/19*	Rectal swab	AM, CS	MEM, GM, CIP, SXT
*A. baumannii 05/19*	Throat swab	AM, CS	MEM, GM, CIP, SXT
*A. baumannii 06/19*	Rectal swab	AM, CS	GM, CIP, SXT
*A. baumannii 07/19*	Throat swab	AM, CS	MEM, GM, CIP, SXT
*K. pneumoniae 08/19*	Throat swab	AM, FOS, CS	AMC, TZP, CTX, CAZ, FEP, ETP, MEM, GM, CIP, SXT
*A. baumannii 09/19*	Bronchial aspirate	AM	MEM, GM, CIP, CS, SXT
*K. pneumoniae 11/19*	Rectal swab	CS, SXT	AMC, TZP, CTX, CAZ, FEP, ETP, MEM, AM, GM, CIP, FOS
*A. baumannii 12/19*	Throat swab	AM, CS	MEM, GM, CIP, SXT
*K. pneumoniae 13/19*	Rectal swab	AM, FOS, CS	AMC, TZP, CTX, CAZ, FEP, ETP, MEM, GM, CIP, SXT
*K. pneumoniae 14/19*	Urinary catheter	FOS, CS	AMC, TZP, CTX, CAZ, FEP, ETP, MEM, AM, GM, CIP, SXT
*K. pneumoniae 18/19*	Rectal swab	MEM, AM, FOS, CS	AMC, TZP, CTX, CAZ, FEP, ETP, GM, CIP, TGC, SXT
*A. baumannii 22/19*	Throat swab	CS	MEM, AM, GM, CIP, SXT
*K. pneumoniae 23/19*	Throat swab	CS	AMC, TZP, CTX, CAZ, FEP, ETP, MEM, AM, GM, CIP, FOS, SXT
*K. pneumoniae 24/19*	Rectal swab	CS	AMC, TZP, CTX, CAZ, FEP, ETP, MEM, AM, GM, CIP, FOS, SXT
*C. albicans 01/18*	Vaginal swab	CAS, MYC, AMB	FLU, VOR
*C. albicans 05/18*	Blood	FLU, VOR, CAS, MYC, AMB	
*C. glabrata 10/19*	Bronchial aspirate	CAS, MYC, AMB	
*C. parapsilosis 16/19*	Urinary catheter	FLU, VOR, CAS, MYC, AMB	
*C. albicans 25/19*	Foot skin swab	FLU, VOR, CAS, MYC, AMB	
*C. albicans 26/19*	Nasal discharge	FLU, VOR, CAS, MYC, AMB	
*C. albicans 27/19*	Wound swab	FLU, VOR, CAS, MYC, AMB	

Amikacin (AM); Amoxicillin—Clavulanic acid (AMC); Amphotericin B (AMB); Benzylpenicillin (P); Caspofungin (CAS); Cefepime (FEP); Cefotaxime (CTX); Cefoxitin (FOX); Ceftazidime (CAZ); Ciprofloxacin (CIP); Clindamycin (CM); Colistin (CS); Ertapenem (ETP); Erythromycin (E); Fluconazole (FLU); Fosfomycin (FOS); Fusidic acid (FA); Gentamicin (GM); Levofloxacin (LVX); Linezolid (LZD); Meropenem (MEM); Mycafungin (MYC); Mupirocin (MUP); Nitrofurantoin (FT); Oxacillin (OX); Piperacillin—Tazobactam (TZP); Rifampicin (RA); Teicoplanin (TEC); Tetracyclin (TE); Tigecycline (TGC); Trimetoprim—Sulfamethoxazole (SXT); Vancomycin (VA); Voriconazole (VOR).

**Table 3 antibiotics-11-00361-t003:** MBC values% *v*/*v* (dilution of distilled extract of bergamot).

ID Strain	MBC	ID Strain	MBC
*S. aureus 3/18*	2.5% *v*/*v* (1:40)	*K. pneumoniae 11/19*	1.25% *v*/*v* (1:80)
*S. aureus 2/18*	2.5% *v*/*v* (1:40)	*A. baumannii 12/19*	1.25% *v*/*v* (1:80)
*S. aureus 1/18*	5% *v*/*v* (1:20)	*K. pneumoniae 13/19*	5% *v*/*v* (1:20)
*S. epidermidis 19/19*	5 % *v*/*v* (1:20)	*K. pneumoniae 14/19*	1.25% *v*/*v* (1:80)
*S. epidermidis 28/19*	5% *v*/*v* (1:20)	*K. pneumoniae 18/19*	2.5% *v*/*v* (1:40)
*A. baumannii 29/19*	1.25% *v*/*v* (1:80)	*A. baumannii 22/19*	0.625% *v*/*v* (1:160)
*A. baumannii 04/18*	1.25% *v*/*v* (1:80)	*K. pneumoniae 23/19*	2.5% *v*/*v* (1:40)
*K. pneumoniae 30/19*	2.5% *v*/*v* (1:40)	*K. pneumoniae 24/19*	2.5% *v*/*v* (1:40)
*K. pneumoniae 02/19*	2.5% *v*/*v* (1:40)	*C. albicans 01/18*	2.5% *v*/*v* (1:40)
*A. baumannii 03/19*	0.625% *v*/*v* (1:160)	*C. albicans 05/18*	2.5% *v*/*v* (1:40)
*A. baumannii 04/19*	1.25% *v*/*v* (1:80)	*C. glabrata 10/19*	0.313% *v*/*v* (1:320)
*A. baumannii 05/19*	2.5% *v*/*v* (1:40)	*C. parapsilosis 16/19*	1.25% *v*/*v* (1:80)
*A. baumannii 06/19*	0.625% *v*/*v* (1:160)	*C. albicans 25/19*	1.25% *v*/*v* (1:80)
*A. baumannii 07/19*	1.25% *v*/*v* (1:80)	*C. albicans 26/19*	1.25% *v*/*v* (1:80)
*K. pneumoniae 08/19*	5% *v*/*v* (1:20)	*C. albicans 27/19*	1.25% *v*/*v* (1:80)
*A. baumannii 09/19*	2.5% *v*/*v* (1:40)		

## Data Availability

The present study includes all available data.

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
