# Peer review of "Citrus bergamia: Kinetics of Antimicrobial Activity on Clinical Isolates"

_antibiotics, 2022, doi:10.3390/antibiotics11030361_

Round 1

Reviewer 1 Report

The article of Quirino A. and co-workers discuss the antimicrobial activity of distilled extract of bergamot (Citrus bergamia) on different pathogenic bacteria and fungal strains of clinical relevance. Although the results are interesting and contribute to future alternatives for developing new phytoterapeutic agents (and counteract to the current multi-drugs resistant microorganisms), some points should be revised, and the paper cannot be published in the current form. Overall, authors should complete missing information on experimental methods, chemical composition of bergamot extract (at least major components), increase number of biological replicates in experimental design, major changes in the context/connection of results and discussion section along with improved presentation of results.

Major comments:
â–ª Improve the introduction section mainly on two themes: 1) the bacterial and fungal species used (S. aureus, K. baumannii, A. baumannii, Candida species) in the study as representative pathogens (impact, treatments, antibiotic-resistance, among others), 2) the essential oil of bergamot (chemical composition and biological activities).
â–ª Some of the material and methods sections have not clear descriptions. Please, check it and improve the grammatical English (please, use correct verb tenses. Line 90 replace “will” by “was”).
â–ª Section 2.1. ¿What was the final volume of distilled extract of bergamot used for this assay, at the concentrations tested? ¿What was the vehicle/carrier used to prepare the distilled extract for this assay? Bacterial suspension without distilled extract (or plant extract) is the control sample, not positive control. Please, check it and adjust the document. On the other hand, I suggest to the authors to perform the experiments with at least three biological replicates; two are insufficient to be considered as reproducible and validated results.
â–ª Title of table 1 should be improved. Furthermore, table should contain more information about the microbial identification test used and the sensitivity/resistant to antibiotics and antifungal (perhaps as supplementary material); it is not clear which yeast is resistant to fluconazole or voriconazole (or both).
â–ª Section 3.1. First paragraph describes the chemical composition of distilled extract of bergamot by GC. It was GC-FID? GC-MS? Why is this information not described in the material and methods section? What was the compound identification and quantification method used?
â–ª Results and discussion section was difficult to understand because not information about the voucher microorganisms were described along with the results obtained. I strongly recommend showing antimicrobial data in a table result. From all tested microorganisms, which were more susceptible to the distilled extract? It is not clear; the information is very general.
â–ª Line 166: please, clarify this discussion of the result with Candida species. Was the lipid composition evaluated for the fungal isolates used in the study? or why the authors suggest that lipid composition of cell membranes was the cause of such observation? Is
there any evidence/support of this finding? This part needs to be check it and expand a bit more the information related there.
â–ª Figure 1: Grey colors used in the figures (A-F) are difficult to associate with specific strain (difficult to read). I recommend using different colors or geometric shapes that improve descriptors. Please, check figure 1.E and correct if it is the case.
â–ª On overall, the sections 3.1 and 3.2 need to be improved, specially, the discussion of the results. Correlation of chemical pattern from plant extract associated with biological activities can be very helpful (supported by the findings of other authors).

Minor comments:
• Line 30: replace saggest by suggest.
• Line 45-46 and 52: repeated information. Please, check and adjust it.
• Line 140. Replace antimicrobic by antimicrobial.
• Line 180-182: why authors compare results with thyme oil? Is this oil similar with bergamot (in chemical composition)?
• Line 211: what means FBC? And what is the connection of this abbreviature with bactericidal (line 209)?
• Figure 4. how can explain the result observed in treated for 0´?

Reviewer 2 Report

This article summarizes the Citrus bergamia: Kinetic of Antimicrobial Activity on Clinical Isolates. This manuscript appears rough and confused. The experimental approach is incomplete. It should be strongly revised before it is considered as a potential publication in Antibiotics. Some examples are listed below

The introduction is quite uneven.

In the Materials and Methods section there are many issue to be solved such as:

No statistical Analysis

No description and identification of plants. Also photo of plant it is must be included.

Table 1 should be reported in the results and should be organized differently. The antimicrobial resistance of the isolates should be reported clearly.

Method of Distilled extraction of Citrus bergamia must be reported.

A complete characterization of the Distilled extract must be reported.

The results and discussion section is approximate, it is must be seperate Results from Discussion.

References Only 22, however I believe the issue of manuscript has been extensively covered in published works.  

Reviewer 3 Report

The manuscript "Citrus bergamia: Kinetic of Antimicrobial Activity on Clinical Isolates" presents interesting results from antimicrobial research and may be revised in the Antibiotics journal

Detailed comments:

Have the isolated strains been deposited in publicly available collections?

Chapter 2.1 - the title of the chapter does not correspond to the content - the MIC was determined using the microwell method, and the MBC using the plate method.

Why was Blood Agar plates selected and not Mueller-Hinton Agar?

Figure 1 - please provide a numerical scale on the OY axes, not a scientific scale, the drawings will be clearer.

Chapter 3.1 - please add a table with MIC and MBC results, for the reader it will be more meaningful. 

Round 2

Reviewer 1 Report

Although I have noticed some changes on the final version of the manuscript from Quirino and co-workers, I think that the structure and content of the discussion section can be improved; there are several lines with general information and, the most important, is missing the connection with the results obtained. With the above, some additional comments are mentioned below:

  • Figure 2. Use italics for species names (please, check the entire manuscript and correct).
  • Section 2.1 contain information does not belong to materials and methods. Part of this info could be included in the discussion section. Please, check and correct.
  • Section 2.3. Correct “olumn” by “column”.
  • Section 2.5. Change “Microwell dilution assay” by “Broth Microdilution assay”.
  • Table 3. Change commas by dots as decimal separator (please, check the entire manuscript and correct).
  • Figure 2. I strongly recommend using software like “origin or origin-Pro” to improve the time-kill curves.
  • Figure 5. Micrography for “treated for 0´” is not clear and does not support the findings during treatment with the essential oil on the timeline analyzed.
  • The conclusion is too general, it should be connected and supported with the results.

Reviewer 2 Report

Thank you for all changes. The paper now totally changed to better. The quality of article now is very good.

Figure 2: Resolution is not good

Figure 5: 30, - 60,.... what is mean comma above numbers?. please corrected it.

Please just check names of microorganisms and changed to italic.

Reviewer 3 Report

The corrections have been made in the text, I have no more comments.